# Plasmids as Key Players in *Acinetobacter* Adaptation

**DOI:** 10.3390/ijms231810893

**Published:** 2022-09-17

**Authors:** Olga Maslova, Sofia Mindlin, Alexey Beletsky, Andrey Mardanov, Mayya Petrova

**Affiliations:** 1Institute of Molecular Genetics of National Research Centre “Kurchatov Institute”, 123182 Moscow, Russia; 2Institute of Bioengineering, Research Centre of Biotechnology of the Russian Academy of Sciences, 119071 Moscow, Russia

**Keywords:** plasmid backbone, accessory region, antibiotic resistance, heavy metals resistance, recombination

## Abstract

This review briefly summarizes the data on the mechanisms of development of the adaptability of *Acinetobacters* to various living conditions in the environment and in the clinic. A comparative analysis of the genomes of free-living and clinical strains of *A. lwoffii*, as well as the genomes of *A. lwoffii* and *A. baumannii,* has been carried out. It has been shown that plasmids, both large and small, play a key role in the formation of the adaptability of *Acinetobacter* to their living conditions. In particular, it has been demonstrated that the plasmids of various strains of *Acinetobacter* differ from each other in their structure and gene composition depending on the lifestyle of their host bacteria. Plasmids of modern strains are enriched with antibiotic-resistant genes, while the content of genes involved in resistance to heavy metals and arsenic is comparable to plasmids from modern and ancient strains. It is concluded that *Acinetobacter* plasmids may ensure the survival of host bacteria under conditions of various types of environmental and clinical stresses. A brief overview of the main mechanisms of horizontal gene transfer on plasmids inherent in *Acinetobacter* strains is also given.

## 1. Introduction

Bacteria of the genus *Acinetobacter* are gram-negative coccobacilli belonging to the family *Moraxellaceae*, class γ-proteobacteria. *Acinetobacters* are aerobic chemoorganotrophic saprophytes. Strains of *Acinetobacter* are among the most ubiquitous bacteria and live in a wide variety of ecological niches. They are widespread in various soils, fresh and sea water, on surface covers and in the intestines of insects, animals and humans, and on plants [1,2]. Some strains, predominantly *A. baumannii*, are causative agents of nosocomial infections such as septicemia, pneumonia, meningitis, urinary tract infections, skin infections, gastroenteritis and wound infections, as well as infectious diseases in animals [3,4]. *A. baumannii* strains are also particularly dangerous because many of them are resistant to all antibacterial drugs used in clinic [5,6]. The number of studies of clinical strains of *Acinetobacter* increases from year to year. One of the most important questions facing researchers is the question about the mechanisms underlying the amazing plasticity of representatives of the genus *Acinetobacter* and their ability to adapt to a wide range of living conditions [7,8,9].

The first attempt to compare the structure of the genomes of different *Acinetobacter* strains was made in 2008 [10]. Three strains were chosen for comparison, two of which belonged to the *A. baumannii* species and one to the *A. baylyi* species. The strain AYE of *A. baumannii* was a human isolate; the strain SDF was isolated from lice, and the strain ADP1 (BD413) of *A. baylyi* was a mutant of the naturally transformable strain BD4 isolated from soil [2,11]. The findings show that the genome structure of the three compared strains is partly determined by life in different ecological niches. In particular, multidrug resistance was found only in the clinical AYE strain [10]. To the surprise of the researchers, the clinical strain had the greatest metabolic potential. However, most of the differences found cannot be unambiguously attributed to adaptation. Perhaps multiple resistance to antibiotics is the only reliable example of adaptation to the conditions of existence in the clinic. Obviously, it is impossible to find a clue to the plasticity of *Acinetobacter* by comparing the genomes of only three strains from two species.

In further studies, chromosomes and plasmids of various *Acinetobacter* species were sequenced and characterized. However, only very limited information about the comparative genomics of strains characterized by different lifestyles could be obtained from these works [9,12].

We had at our disposal a collection of ancient strains of *Acinetobacter* isolated from the permafrost of the Kolyma Lowland, which guarantees that these strains had never experienced anthropogenic influence. Five strains of *A. lwoffii* isolated from permafrost samples aged from 15 thousand to 1.8 million years were selected from this collection for the comparative analysis of genomes. Representatives of this species are known to exist both in the environment and on human skin and mucous membrane [13]. In addition, strains of this species have been increasingly reported as hospital pathogens associated with bacteremia [14]. We sequenced the genomes of these strains and performed a comparative study of plasmids and chromosomes of modern and ancient strains.

Thanks to these studies, we were able to get an idea of how rapid adaptation to habitat conditions is ensured in representatives of the genus *Acinetobacter*. In this minireview, we summarize our findings and recently published results of the other authors and try to answer the question about the role of *Acinetobacter* plasmids in shaping the adaptability of host bacteria in different habitats.

## 2. Results

### 2.1. Comparison of the Chromosomes of A. lwoffii

To reveal specific genetic characteristics of ancient *A. lwoffii* strains preserved in permafrost and their differences from modern isolates, we conducted a comparative analysis of the genome structure of the five permafrost *A. lwoffii* strains mentioned above (Table 1) and modern environmental and clinical strains of this species (Appendix A). It should be noted that analysis of the genomes of strains isolated from permafrost provides a unique opportunity to study microorganisms that existed before the onset of anthropogenic impact on the biosphere [15,16]. All strains were resistant to various salts of heavy metals and to some antibiotics (Table 1). To compare them with modern strains, we reiterated our previous analysis of *A. lwoffii* chromosomal sequences using all available genomes of *A. lwoffii* placed in the database by 1 July 2022. For the construction of a phylogenetic tree, we identified a set of 847 single-copy genes present in all genomes using the CD-HIT v4.6 clustering program with a 90% nucleotide global identity threshold. Individual alignment for each single copy gene nucleotide sequences was made using MAFFT v7.453; the alignments of 847 genes were concatenated and used as an input for the tree construction in PhyML v3.3 with default parameters. As a result of this analysis, it turned out that five of the strains retrieved from the database do not belong to the species *A*. *lwoffii* (Appendix A). We therefore removed them from the analysis and conducted it again without their genomes. The resulting tree is shown in Figure 1. 

Surprisingly, we did not find any significant differences in the structure (sequences) of chromosomes that would distinguish permafrost strans from modern strains, including clinical ones [17]. It is clearly seen that there are no isolated groups of *A. lwoffii* strains adapted to any habitat. Moreover, phylogenetic analysis based on the whole genome sequences showed that permafrost (ancient) strains do not form a distinct cluster, and some of them are closely related to clinical isolates.

### 2.2. Comparison of the Plasmids of A. lwoffii

Since comparison of the complete genomes of *A. lwoffii* strains did not reveal significant differences between clinical and environmental, or modern and permafrost isolates, we decided to compare the plasmids of modern and ancient strains of *A. lwoffii.* In particular, we analyzed the content of heavy metal and antibiotic-resistant genes in them. For this analysis, the sequences of all *A. lwoffii* plasmids placed in the database by 1 July 2022 were used. We compiled a detailed list of all resistant genes found in the plasmids of modern and ancient *A. lwoffii* strains (Appendix A) and determined their average numbers per plasmid (Table 2). In this analysis, the presence of only complete operons (genes) of resistance was taken into account. The results show that modern plasmids are significantly enriched with antibiotic-resistant genes compared to ancient ones, while the content of genes for resistance to heavy metals and arsenic is comparable (Table 2). It should be emphasized that strains of *A. lwoffii*, unlike *A. baumannii*, are part of the normal microbiota of the skin and mucous membranes of healthy people [13]; therefore, it is likely that many strains isolated from human samples are not pathogenic. However, modern strains of *A. lwoffii* likely begin to accumulate the corresponding resistant genes when exposed to antibiotics, which is typical for clinical and veterinary specimens. It should be noted that the accumulation of new adaptive genes in *A. lwoffii* also occurs mainly on plasmids [17].

Many plasmids of modern *A. lwoffii* strains contain remnants of different heavy metals and arsenic resistance operons. For example, two variants of the *ars*-operon with an incomplete set of genes were found: (i) an operon contained a single *arsH* gene in which the N-terminus was deleted; and (ii) an operon that contained three genes, including *arsH,* in which the N-terminus was deleted; the repressor gene, and *arsC.* In both cases, the defective operon contained an IS element next to the *arsH* gene: IS*1007* in the operon of the first type and IS*66* in the second. Most likely, the formation of defective *ars*-operons is the result of the insertion of the IS element into the N terminus of the *arsH* gene. At the same time, only complete *ars*-operons were found in ancient bacterial strains [16].

Interestingly, the genomes of ancient permafrost strains of *A. lwoffii* and modern environmental strains (Appendix A) are more similar in terms of the number of plasmids and the content of heavy metal and antibiotic resistance genes in them. For instance, the plasmidome generated from an arsenic-resistant strain, *A. lwoffii* ZS207, contained nine plasmids in the size between 4.3 and 38.4 kb as well as one 186.6 kb megaplasmid. The mega-plasmid carries arsenic and heavy metals-resistant regions similar to those found in permafrost *A. lwoffii* strains [18]. Numerous plasmids (15) were also found in the strain *A. lwoffii* M2a isolated from a honey sample [19]. Although some of them were unassembled, it was possible to show that they contain genes of mercury and heavy metal salts resistance; antibiotic-resistant genes were not found in these plasmids [19]. The same pattern was observed in unassembled plasmids of the *A. lwoffii* strain GT2 [20,21]. These results show that both ancient and modern strains of *A. lwoffii* usually carry many plasmids, and these plasmids, especially large ones, usually contain different sets of genes for resistance to heavy metals and arsenic. At the same time, plasmids from environmental strains of *A. lwoffii* rarely bear antibiotic-resistant genes, in contrast to the strains isolated in the clinic (which prevail in the sequenced genomes of *A. lwoffii*, Table 2).

### 2.3. Comparative Structure of A. lwoffii and A. baumannii Genomes

It was interesting to compare the genomes of *A. lwoffii* and *A. baumannii* strains, since there are much more pathogenic strains among *A. baumannii*, and most of the sequences in the database belong to clinical isolates. A comparative analysis of the genomes of *A. lwoffii* and *A. baumannii* strains revealed a number of differences between them: (i) chromosome sizes in *A. baumannii* are larger than in *A. lwoffii* by about 20%; (ii) in contrast, the number of plasmids and their total size are greater in *A. lwoffii* than in *A. baumannii*; (iii) environmental strains of *A. lwoffii* outnumber *A. baumannii* strains in the number and diversity of heavy metal-resistant genes, and these genes are predominantly located on plasmids in *A. lwoffii* and in the chromosomes in *A. baumannii* (Table 3) [16,17].

In addition, it is known that chromosomes of *A. baumannii* can contain resistance and pathogenicity islands, the acquisition of which immediately gives many advantages to the strain living in the clinic [22,23,24]. These structures have not yet been found in strains of *A. lwoffii* [22]. Thus, different *Acinetobacter* species may have slightly different adaptation strategies associated with different contributions of plasmids and chromosomes to them. Analysis of available data shows that the genes of resistance to heavy metals are markers of adaptation to the environment and the genes of resistance to antibiotics are characteristics for clinical strains of *Acinetobacter*. Furthermore, it was demonstrated that in some species of *Acinetobacter*, such as *A. lwoffii,* plasmids are the main players in adaptation, while in other species, such as *A. baumannii*, their role is apparently less important, but still significant [7,16,25,26,27,28,29].

### 2.4. Acquisition of New Genes in Small Plasmids of Acinetobacter

Comparative analysis of one of the most abundant groups of small plasmids, the representatives of which are very often found in both clinical and environmental strains of different species of *Acinetobacter*, allowed us to see how the formation of new variants occurs. We demonstrated that all members of this group have similar genes of plasmid maintenance (the backbone region), but they strongly differ in the structure of their accessory regions (Figure 2) [30]. Variants of the plasmid pRAY* with the *aadB* gene for resistance to kanamycin and gentamicin are most common among modern strains [31]. Three out of the five permafrost strains of *A. lwoffii* contain a related plasmid, pALWED1.8, about 4.1 kb in size with the *aad27* gene for streptomycin and spectinomycin resistance. An almost identical plasmid was also found in modern *Acinetobacter* strains of different species [30]. All other related plasmids are arranged in a similar way (Figure 2). Another very common variant is a plasmid with detergent-resistant gene encoding alkyl/arylsulfatase [30]. However, there are many other rare variants that contain ORFs encoding hypothetical proteins with obscure functions or that do not contain any coding sequences. These sequences most often resemble regions of *Acinetobacter* chromosomes [30]. It can be proposed that small pieces of DNA are constantly being inserted into these plasmids, followed by their selection and wide distribution of most adaptive variants. However, the mechanism of active DNA acquisition by these plasmids is unclear.

### 2.5. Acquisition of New Genes by Acinetobacter Mega-Plasmids

Very rapid DNA rearrangements and acquisition of new adaptive genes were also revealed in comparative analysis of the sequences of mega-plasmids. Two groups of researchers identified a new group of giant conjugative mega-plasmids about 300 kb in size and demonstrated their wide distribution in the environmental and clinical *Acinetobacter* strains and their participation in the horizontal transfer of antibiotic resistance genes [32,33,34]. All members of this very large group contain a common backbone region (almost 100 kb in size), which includes genes for replication, plasmid maintenance and conjugative transfer. At the same time, analysis of their accessory regions demonstrates high abundance of various resistance genes. In total, 221 different antibiotic-resistant genes were revealed in 21 mega-plasmids from clinical strains of *Acinetobacter* analyzed by Ghaly et al. [32]. In the study of Mindlin et al. [34], the difference in the structure of clinical and natural variants of mega-plasmids was demonstrated by the example of pALWED1.1 from an environmental permafrost strain of *A. lwoffii* and pAHTJR1 from a clinical *A. haemolyticus* strain. The permafrost plasmid contains only one antibiotic resistance gene, *tet* (*H*), with impaired function, and a large cluster of determinants of resistance to heavy metals include mercury, chromium, cobalt, zinc, cadmium and nickel. These genes are absent in the clinical version of the plasmid, as well as some other genes not associated with antibiotic resistance. At the same time, the clinical plasmid contains three regions with genes for resistance to different antibiotics.

For this review, we analyzed the content of determinants of resistance to heavy metals, arsenic and antibiotics in 36 mega-plasmids from *Acinetobacter* strains of various species and different origins (Figure 3). Antibiotic-resistant genes were identified with AMRFinderPlus v3.10 using default parameters. Metal-resistant genes were identified using a diamond v2.0.13.151 homology search against the BacMet2 database; alignment coverage of at least 80% of the reference gene and e-value of <1e-5 were used as cutoff parameters. For *czc* and *cop* operons, we additionally used manual selection of reference genes for the diamond search.

This analysis shows that mega-plasmids isolated from strains related to the clinic or veterinary, including *A. baumannii* strains, are very rich in genes resistant to various antibiotics compared to plasmids from strains isolated from the environment. However, it should be noted that of the 36 plasmids whose complete sequences are available in the database, only 4 were isolated from environmental samples, 1 from a permafrost strain and 3 from strains isolated from marine king prawns. At the same time, many clinical plasmids contain certain operons of resistance to heavy metals and/or their relics. Interestingly, mercury-resistant operons are most often found in mega-plasmids. This may be due to the fact that mercury preparations had been used in medicine for a long time. At the same time, no complete operon of resistance to arsenic was found in mega-plasmids. Overall, analysis of mega-plasmids from predominantly clinical strains of *Acinetobacter* of various species reveals similar trends to the analysis of plasmids from *A. lwoffii*. 

In large plasmids, in contrast to small ones, several mechanisms of acquisition of resistant genes are evident, including insertions of composite transposons and various integrons [34]. The latter are very common and are found in at least half of modern mega-plasmids. Moreover, some mega-plasmids contain 2–3 integrons [34]. Integrons are quite rare in unspoiled ecosystems but are abundant in the clinic, where they encode resistance to antibiotics [35,36,37]. The transfer of integrons mainly occurs through their insertion into both simple and compound transposons [38]. Similarly, compound transposons are constantly found in the clinic, while they are practically absent in environmental strains [39,40,41].

### 2.6. Mobile Genetic Elements and Mechanisms of Rearrangement of Acinetobacter Plasmids

Quite a long time ago, it was found that, unlike in other bacteria, mercury-resistant genes in *Acinetobacters* most often are part of defective transposons, with transposition genes almost completely lost [42]. In other bacteria, the loss of mobility of a mercury transposon usually leads to its elimination from cells. On the contrary, such defective transposons are widely distributed on different plasmids in *Acinetobacter* strains of various species [42]. Detailed analysis of the structure of defective *mer* transposons demonstrated that their distribution mainly depends on various systems of homologous and site-specific recombination. IS elements often act as homologous sites for recombination events. On the basis of DNA sequence analysis, possible mechanisms of translocation of defective mercury-resistant transposons via recombination events implicating nearby *res* (resolution) sites and IS elements were proposed (Figure 4) [42]. In this work, evidence was also obtained for an unusually active rearrangement of *Acinetobacter* plasmids, since at least 36 recombination events over 70 kb of sequenced plasmid DNA were documented [42]. In addition, a site-specific recombination system CinH-RS2, encoding a 189 aa CinH recombinase and a 119 bp recombination site RS2, is often found in *Acinetobacter* plasmids, including pKLH2, pKLH204 and pKLH205, [43,44,45]. It was shown that this recombination system recognizes not only its own *res* sites, but also the *res* sites of the Tn*3* family transposons, as well as *res* sites of some plasmid cointegrates [43].

Recently, a novel class of mobile genetic elements, the transposition of which likely depends on the action of the *dif/*Xer recombination system, was discovered in *Acinetobacter* species [46]. 

In other bacteria, the recombination site, called the *dif* site is located only in the chromosome, and the tyrosine recombinase XerC/XerD participates in the resolution of chromosomal dimers after replication [47,48]. It turned out that in *Acinetobacter, dif-*like sites (p*dif*) are often found in plasmids [28,49,50,51]. Genes located between two plasmid p*dif* sites can be transferred between plasmids, since identical genes have been found in different plasmids [49,50,52]. Although this has not yet been experimentally proven, it is believed that this transfer involves the same XerC and XerD proteins that recognize the chromosomal *dif* site [46,50,51]. This gene transfer system is likely active in plasmids from both environmental and clinical strains, although the genes in these two cases are different. In plasmids from clinical strains, these genes predominantly encode resistance to antibiotics [49,50,53,54,55,56], whereas in environmental strains, various genes are found, including those that are associated with adaptation to their habitats [50,51]. 

## 3. Conclusions

Among the members of the genus *Acinetobacter*, *A. baumannii* is undoubtedly the most actively investigated and therefore the best studied. The important role of plasmids in the acquisition of unprecedented multidrug resistance by clinical strains of this species has been confirmed by a huge number of studies conducted around the world [8,12,56,57,58,59]. However, little was known about the significance of plasmids in other species of the genus, since nonclinical strains were rarely studied. Our studies of *Acinetobacter* strains isolated from permafrost have shown that in environmental strains, the contribution of plasmids to the adaptation to living conditions may be even higher than in clinical strains. Analysis of available data allows us to conclude that, to achieve rapid exchange of plasmid genes, *Acinetobacters* uses not only the mechanisms common in other bacteria but also an arsenal of their own. In particular, unlike other bacteria, horizontal transfer of mercury resistance operons on *Acinetobacter* plasmids is carried out not by the transposition of functionally active mercury transposons, but by various recombination events of aberrant, and therefore functionally inactive, mercury transposons [42]. Another example is the use of the *dif/*Xer site-specific recombination system, which in other bacteria functions only on chromosomes, to disseminate adaptive plasmid genes [46]. It seems that the adaptation strategy of *Acinetobacter* spp. is to constantly pass different genes through their plastidome. Hernández-González et al. [60] described horizontal gene transfer as “gene flow”. The term flow seems to most clearly reflect the essence of what happens in the plasmids of *Acinetobacter*. Considering that many *Acinetobacter* plasmids are mobilizable, and widespread mega-plasmids are able not only to move themselves but also to mobilize other plasmids [34], it is clear that the necessary genes easily become available to all *Acinetobacter* strains. Then, there is a selection of options to be adapted depending on the given living conditions.

Comparative analysis of the nucleotide sequences of more than 40 plasmids of permafrost strains of *A. lwoffii* and 75 plasmids of modern strains of *A. lwofii* made it possible to trace how the set of genes changes in accessory regions. It was also possible to show that accumulation of antibiotic resistance genes in modern strains occurs not only in modern clinical but also in modern environmental strains, indicating a strong anthropogenic impact on the biosphere associated with the production and widespread use of antibiotics over the past seventy years.

Thus, we come to the conclusion that the adaptability of *Acinetobacter* is provided not only by their high metabolic potential encoded in the chromosome, but to a large extent by their plasmids, although this contribution can obviously be different in different species.

## Figures and Tables

**Figure 1 ijms-23-10893-f001:**
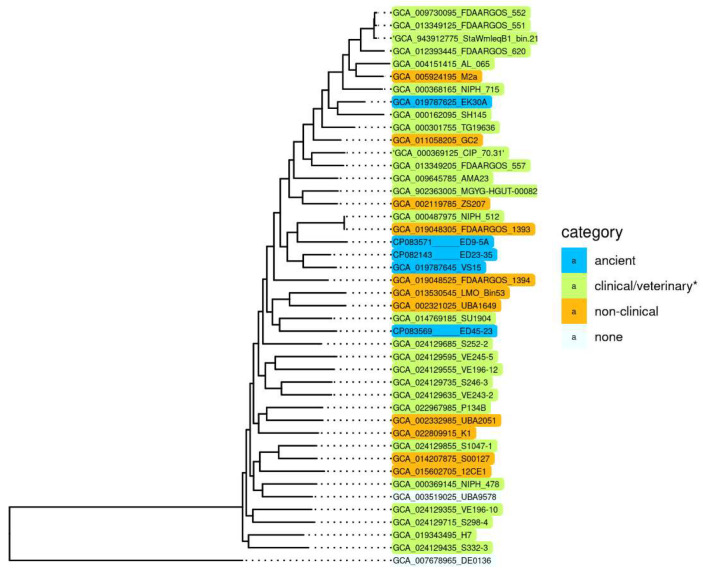
Genome-based phylogeny of *A. lwoffii*. The maximum likelihood tree is based on concatenated nucleotide sequences of 847 single copy genes. * Strains isolated from samples of healthy and sick animals and humans. *A. pseudolwoffii* DE0136 was used to root the tree. The bootstrap values for all nodes are above 99%.

**Figure 2 ijms-23-10893-f002:**
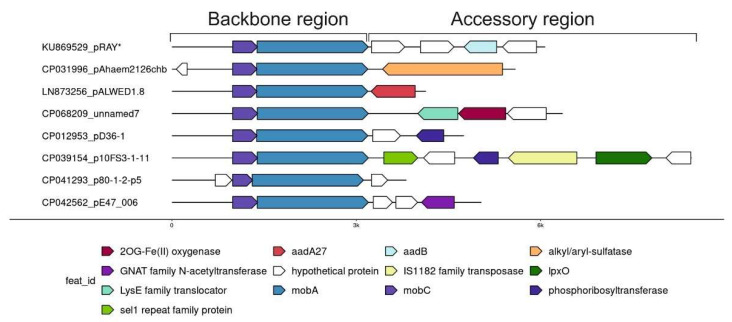
Plasmid pRAY* and related small plasmids.

**Figure 3 ijms-23-10893-f003:**
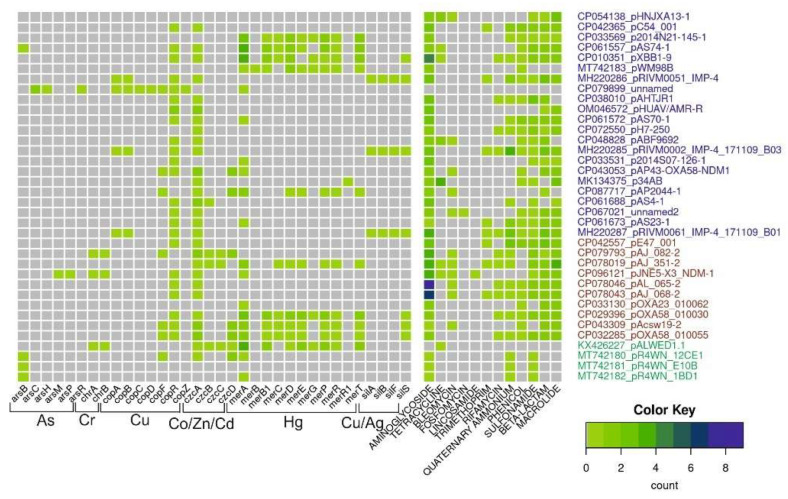
Diversity and abundance of heavy metals and antibiotic-resistant genes in mega-plasmids. The number of resistant genes (color scale bar) is shown for each plasmid (rows). Plasmid names are colored according to their origin: from Homo sapiens, pets, poultry, in blue; from hospital and farm environments, including hospital sewage in red; from the environment in green.

**Figure 4 ijms-23-10893-f004:**
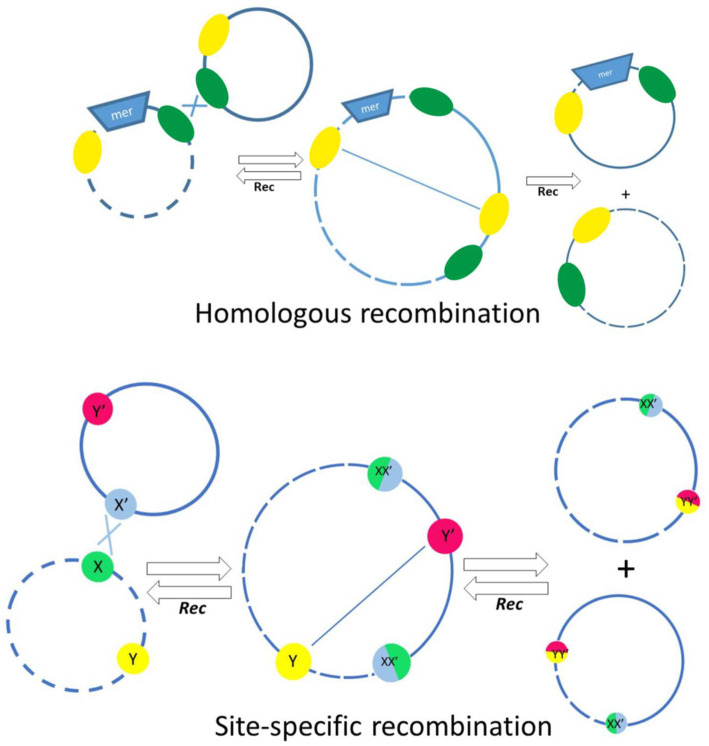
The main mechanisms involved in the exchange of genetic material between plasmids, involving homologous recombination (**top**) or site-specific recombination depending on *res* sites and the action of resolvases (**bottom**). The *res* sites are indicated by colored circles in the bottom panel.

**Table 1 ijms-23-10893-t001:** Analyzed permafrost strains of *A. lwoffii*.

Strain	Isolation Depth (m)	Age of Permafrost (Thousand Years)	Resistance to Antibiotics	Resistance to Heavy Metals	Accession Number
ED25-35	4.5	20–40	Sm, Sp	Hg, Cr, Co, Cd, Zn, Ni	CP082143.1
ED45-23	2.9	20–40	-	Hg, As, Cu	CP083569.1
ED9-5A	6.5	15–30	-	Hg, As, Cr, Cd, Zn, Cu	CP083571.1
VS15	34.0	20–40	Ap, Cm, Sm, Sp	Co, Cd, Zn, Cu	CP080576.1
EK30A	47.9	1600–1800	Ap, Sm, Sp	Cr, Co, Cd, Cu	CP080636.1

**Table 2 ijms-23-10893-t002:** The number of plasmid-encoded heavy metal and antibiotic-resistant genes (operons) in permafrost and modern *A. lwoffii* strains.

Group of Plasmids	Plasmid Number	Number of Genes (Determinants) of			
		Heavy Metal Resistance		Antibiotic Resistance	
		Total	Per Plasmid	Total	Per Plasmid
From permafrost strains	41	30	30/41 = 0.7	3	3/41 = 0.07
From modern strains	75	33	33/75 = 0.44	41	41/75 = 0.55

**Table 3 ijms-23-10893-t003:** Comparative structure of *A. lwoffii* and *A. baumannii* genomes [17].

Species	Genome Size	Plasmid Number	Content of Plasmid DNA, % of the Genome	Location of Heavy Metal Resistance Genes
*A. lwoffii*	3,408,464	5–15	7.5%	mainly in plasmids
*A. baumannii*	4,018,426	0–4	2.0%	mainly in the chromosome

## Data Availability

Not applicable.

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
