# Peer review of "Plasmids as Key Players in Acinetobacter Adaptation"

_ijms, 2022, doi:10.3390/ijms231810893_

Round 1

Reviewer 1 Report

The authors describe the role of plasmids in Acinetobacter spp (A.lwoffii and A. baumannii) adaption on different environments (mainly in nature and in hospital settings).  They present their own experimental work as well as reviewing existing literature. The manuscript is quite interesting.

Comments

·         Line 26-27:” Bacteria of the genus Acinetobacter represent gram-negative coccobacilli that belong to the family Moraxellaceae, a class of γ-proteobacteria. Acinetobacter spp are aerobic chemoorganotrophic..” instead of “Bacteria of the genus Acinetobacter, gram-negative coccobacilli, belong to the family Moraxellaceae, a class of γ-proteobacteria. Acinetobacter’s are aerobic chemoorganotrophic”

·         Line 54: delete:” a great many”

·         Line 62: “exist” instead of “to have both”

·         Line 100: “five” instead of “5”

·         Line 280:”Acinetobacter” instead of “Acinetobacters”

·         Line 304: “Moreover, genes” instead of “And genes “

·         Line 309-312: “although the genes in these two cases are different. In plasmids from clinical strains, these genes encode resistance to antibiotics [49, 50, 53, 54, 55] whereas, in environmental strains , various genes are encountered, including those associated with adaptation to their habitats [50, 51].” instead of “only the genes in these two cases are different. In plasmids from clinical strains -  these are genes encoding resistance to antibiotics [49, 50, 53, 54, 55] and in environmental  strains - various genes including those that associated with adaptation to their habitats 311 [50, 51].”

·         Line 315: “in order to achieve rapid exchange” instead of “for the rapid exchange”

·         Line 322: “Acinetobacter spp” instead of “Acinetobacters”

·         Line 329: “Then, there is a selection of options to be adapted depending on given living conditions” instead of “And then there is a selection of options that are most adapted to  given living conditions”

·         I would recommend incorporating the following references:

1.      Ibrahim S, Al-Saryi N, Al-Kadmy IMS, Aziz SN. Multidrug-resistant Acinetobacter baumannii as an emerging concern in hospitals. Mol Biol Rep. 2021 Oct;48(10):6987-6998. doi: 10.1007/s11033-021-06690-6.

2.      Liu H, Moran RA, Chen Y, Doughty EL, Hua X, Jiang Y, Xu Q, Zhang L, Blair JMA, McNally A, van Schaik W, Yu Y. Transferable Acinetobacter baumannii plasmid pDETAB2 encodes OXA-58 and NDM-1 and represents a new class of antibiotic resistance plasmids. J Antimicrob Chemother. 2021

3.      Wang Z, Li H, Zhang J, Wang X, Zhang Y, Wang H. Identification of a novel plasmid-mediated tigecycline resistance-related gene, tet(Y), in Acinetobacter baumannii. J Antimicrob Chemother. 2021

4.      Hamidian M, Ambrose SJ, Blackwell GA, Nigro SJ, Hall RM. An outbreak of multiply antibiotic-resistant ST49:ST128:KL11:OCL8 Acinetobacter baumannii isolates at a Sydney hospital. J Antimicrob Chemother. 2021

5.      Frenk S, Temkin E, Lurie-Weinberger MN, Keren-Paz A, Rov R, Rakovitsky N, Wullfhart L, Nutman A, Daikos GL, Skiada A, Durante-Mangoni E, Dishon Benattar Y, Bitterman R, Yahav D, Daitch V, Bernardo M, Iossa D, Zusman O, Friberg LE, Mouton JW, Theuretzbacher U, Leibovici L, Geffen Y, Gershon R, Paul M, Carmeli Y. Large-scale WGS of carbapenem-resistant Acinetobacter baumannii isolates reveals patterns of dissemination of ST clades associated with antibiotic resistance. J Antimicrob Chemother. 2022

Author Response

Thank you very much for your attentive and thoughtful reading of our manuscript and very useful comments and corrections.

We fully agree with all your comments and suggestions and have corrected the text accordingly. We  included all the articles you suggested, except for the last reference, because in our opinion it has no direct relation to the topic of our review. We have also edited the English, according his suggestion. All changes you can track in the edited version of the manuscript. Thanks again.

Reviewer 2 Report

In this review, the authors have discussed out the role of plasmids in pathogenicity of Acinetobacter. In order to validate their hypothesis, the authors have compared the pathogenic strain of the bacteria along with its non-pathogenic counterpart and more interestingly to the ancient bacterial strains that were preserved from permafrost. The authors found out that like many other pathogens, the horizontal gene transfer played a key role in acquisition of genes that conferred antibiotic resistance to the pathogenic strain that is missing from its non-pathogenic cousin. Overall, the review is well written and should be of interest to readers involved in studying bacterial pathogenesis as well as evolution.

Author Response

Thank you very much for your interest in our work and its high appreciation.